# Different Types of Glucocorticoids to Evaluate Stress and Welfare in Animals and Humans: General Concepts and Examples of Combined Use

**DOI:** 10.3390/metabo13010106

**Published:** 2023-01-09

**Authors:** María Botía, Damián Escribano, Silvia Martínez-Subiela, Asta Tvarijonaviciute, Fernando Tecles, Marina López-Arjona, José J. Cerón

**Affiliations:** 1Interdisciplinary Laboratory of Clinical Analysis, Interlab-UMU, Regional Campus of International Excellence Campus Mare Nostrum, University of Murcia, 30100 Murcia, Spain; 2Department of Animal Production, Veterinary School, Regional Campus of International Excellence Campus Mare Nostrum, University of Murcia, 30100 Murcia, Spain

**Keywords:** glucocorticoids, cortisol, cortisone, corticosterone, measurement, stress

## Abstract

The main glucocorticoids involved in the stress response are cortisol and cortisone in most mammals and corticosterone in birds and rodents. Therefore, these analytes are currently the biomarkers more frequently used to evaluate the physiological response to a stressful situation. In addition, “total glucocorticoids”, which refers to the quantification of various glucocorticoids by immunoassays showing cross-reactivity with different types of glucocorticoids or related metabolites, can be measured. In this review, we describe the characteristics of the main glucocorticoids used to assess stress, as well as the main techniques and samples used for their quantification. In addition, we analyse the studies where at least two of the main glucocorticoids were measured in combination. Overall, this review points out the different behaviours of the main glucocorticoids, depending on the animal species and stressful stimuli, and shows the potential advantages that the measurement of at least two different glucocorticoid types can have for evaluating welfare.

## 1. Introduction

Currently, stress is defined as a state of threat to homeostasis [1]. Glucocorticoids are presently the group of biomarkers most frequently used to evaluate the physiological response to stress [2]. The reason is that from a neuroendocrinological point of view, any stressful stimulus triggers the release of the adrenocorticotropic hormone (ACTH), which leads to the secretion of these molecules (Figure 1) [3,4,5,6].

The most common glucocorticoid used to assess stress in humans and many animal species is cortisol [7]; although others such as cortisone [8] and corticosterone (this last one in species such as rats and mice, birds, and reptiles) [9,10] can also be measured. In addition, another way to assess the activity of the hypothalamic–pituitary–adrenal (HPA) axis in stressful situations is via the determination of “total glucocorticoids”. The term “total glucocorticoids” refers to what is measured when immunoassays with non-specific antibodies showing cross-reactivity with different glucocorticoids or related metabolites are employed [11].

This review has two main aims. The first one is to provide some general concepts on glucocorticoids, with a special focus on the general characteristics of the main types of glucocorticoids (cortisol, cortisone, and corticosterone), and on the assays and sample types used for their measurement. The second one is to perform a comparative analysis of the studies published in which at least two different types of glucocorticoids (cortisol, cortisone, corticosterone, or total glucocorticoids) were measured in combination. The study of the reports in which the combined use of two or more different types of glucocorticoids is performed is the main novel point of this review, which, overall, will contribute to a better understanding of the different glucocorticoids that can be used to evaluate stress and welfare (understanding welfare as the presence of normal biological functioning and an adequate emotional state [12]) and the possibilities of their combined use.

## 2. General Characteristics of the Main Glucocorticoids

Glucocorticoids are a group of endogenous adrenal hormones with a 21-carbon skeleton that are derived from cholesterol and that are released in a stressful situation. When released, they bind mainly to the corticosteroid-binding globulin (CBG), making them available for use at systemic or tissue level [13]. Their function is performed by intracellular binding to glucocorticoid receptors (GRs), which belong to the family of nuclear receptors [14]. Although the name “glucocorticoids” originates from their effects on plasma glucose, they are also involved in catabolic metabolism, inflammatory and immune response, and other physiological functions [15,16].

The main glucocorticoids involved in the stress response are cortisol, cortisone, and corticosterone (Table 1). Their concentrations allow the species to be classified as cortisol-dominant (most mammals) or corticosterone-dominant (such as rats, mice, birds or reptiles). Cortisone is produced mainly in the cortisol-dominant species, and its concentration depends on the activity of the 11β-hydroxysteroid dehydrogenase (11β-HSD) type 2 enzyme, which is expressed mainly in kidney, colon, and salivary glands [14].

In addition, glucocorticoid metabolites derived from 5α- or 5β-reductions, hydroxylation, or reductions of the functional group, such as 11β-hydroxyaetiocholanolone, 11-oxoaetiocholanolone I and II, and 5α-pregnane-3β,11β,21-triol-20-one [30,31], can be measured. These are usually analysed in faeces [32,33,34] because of the variety of glucocorticoid-related metabolites present in them. In this line, the term “faecal corticoid metabolites” (fGCM) instead of “faecal total glucocorticoids” has been used since there are metabolites present in the faeces that can also potentially be measured [29]. 

## 3. Measurement

In general, there are two types of assays for the quantification of glucocorticoids:(1)Those using techniques based on the reaction of an antibody with the analyte to be measured, such as radioimmunoassay (RIA), enzyme immunoassay (EIA), chemiluminescence, and, more recently, bead-based luminescent amplification assays (AlphaLISA). RIA assays are currently used with less frequency due to the need of special facilities and the radioactive nature of some components.(2)Techniques based on the direct quantification of the analyte, including high-performance liquid chromatography (HPLC) [35,36] and liquid chromatography–mass spectrometry (LC-MS/MS) [37,38], with the latter being the most sensitive [8].

The main methods, with some selected references as examples of applications, used for the measurement of each type of glucocorticoid are listed in Table 2.

Glucocorticoids can be measured in different sample types. Although blood has been traditionally frequently used, the stress that individuals suffer from blood collection [29] can interfere with the results. In this line, non-invasive alternatives, such as saliva, hair, faeces, or feathers, are becoming increasingly important [40,59,60,61].

## 4. Studies Where Cortisol and Cortisone Were Measured in Combination

### 4.1. Studies on Animals

#### 4.1.1. Studies on Pigs

Cortisol and cortisone were measured in blood using LC-MS/MS to determine the effect of minimally invasive heart catheterization on animal welfare [62]. For both analytes, a significant increase in basal levels after catheterization was observed, although this increase was greater in cortisone (10-fold) than in cortisol (1.5-fold). This increase occurs earlier in cortisol values; however, cortisone levels remain elevated for a longer time. For this reason, the author considers that measuring both glucocorticoids is important for a better interpretation of the results.

In another report, cortisol and cortisone were measured in plasma and also in the saliva of pigs using LC-MS/MS [63]. This study had a control group and an experimental group that was subjected to a stressor (nasal snare), and samples were obtained at baseline, directly after stress and 30 min after the stimulus [63]. The experimental group showed significantly higher concentrations of both cortisol and cortisone than the control group in saliva samples, with this difference being maintained at 30 min in the case of cortisol. These differences before and after stress in both glucocorticoids were smaller in plasma than in saliva, which the author relates to the stress in the control group caused during blood collection. Both plasma and saliva cortisol concentrations were higher than cortisone concentrations. This, together with the lower variability of results, makes cortisol more reliable than cortisone for these authors.

Both analytes were also measured in pig hair at different reproductive periods using AlphaLISA technology [44]. Cortisone concentrations and the cortisone/cortisol ratio increased to a greater extent than cortisol during periods of higher stress due to an increase in the activity of 11β-HSD type 2. The authors recommend measuring both analytes—cortisol and cortisone—because this allows estimating the activity of 11β-HSD type 2, which was the most sensitive marker to detect chronic stress in their experimental conditions.

#### 4.1.2. Studies on other Species

Studies were carried out on captive-bred rainbow trout, where plasma and water-released cortisol and cortisone were measured by RIA and LC-MS/MS, after the trouts were suspended in the air for 1.5–3 min as a stressor [64,65]. In both plasma and water, cortisol and cortisone levels increased significantly two hours after stressor application, with cortisol showing higher values than cortisone. This makes cortisol a more reliable biomarker according to the authors.

Cortisol and cortisone were also assessed in sheep hair by EIA after bacterial inoculation of the right foot as a model of chronic stress [66]. Samples were taken one week before and three weeks after inoculation. Overall, hair cortisone levels were higher than hair cortisol. Furthermore, cortisone levels increased significantly two weeks after inoculation and cortisol levels decreased from baseline. According to the authors, these results may be due to an increase in the local action of the enzyme 11β-HSD type 2 as a result of stress.

### 4.2. Studies on Humans

In humans, there are more studies than on animals in which cortisol and cortisone are measured. These studies could be divided into those assessing acute stress, those assessing chronic stress, and those that studied selected diseases.

To evaluate acute stress, cortisol and cortisone were measured by LC-MS/MS in the saliva and serum of healthy men, with one group undergoing a stressful psychophysiological situation (Trier Social Stress Test, TSST) and a control group [67]. In this report, salivary cortisone was considered a promising stress marker since, after application of the TSST, it was significantly higher than cortisol levels, possibly due to its rapid generation from cortisol by the action of the 11B-HSD type 2 enzyme. Moreover, salivary cortisone correlated better with serum-free cortisol and other stressor parameters (anxiety and heart rate) than salivary cortisol, in line with other studies [68,69].

Cortisol and cortisone concentrations were also compared in the saliva and plasma of subjects undergoing intense physical exercise using an immunoassay and a chemiluminescence-based assay, respectively [70]. A baseline sample was taken and samples at 5 and 20 min after exercise in the morning and afternoon were taken. Samples were also obtained the following day at the same time but without exercise, serving as the control. Plasma and salivary cortisol and cortisone showed different release patterns throughout the day due to circadian rhythms. In general, salivary cortisol increased more and this increase was maintained longer than salivary cortisone, although the cortisone concentration was higher than the cortisol concentration. In plasma, the increase in both analytes was similar but lower than in saliva. 

To evaluate chronic stress, hair cortisol and cortisone values were measured by LC/MS [71], HPLC [49], and LC-MS [72] in people with emotional and work-related stress and pregnancy status. All three studies had similar results, showing that cortisone was the metabolite showing major increases under the effect of the long-term stressor. This may be due to an increase in 11B-HSD type 2 associated with the chronic stress [49].

Cortisol and cortisone were also used to study diseases such as Cushing’s syndrome [73,74], metabolic syndrome [75], or obesity [76]. Both analytes were increased in these situations, having a similar value for assessing these diseases.

As can be observed, both glucocorticoids were measured in different sample types. Interestingly, cortisol and cortisone correlations between saliva and hair were studied using LC-MS/MS [77]. In this report, saliva samples of female students were collected on three consecutive weekends, while a single hair sample was taken two weeks after the last saliva collection. The results determined that there was a good correlation when hair cortisol and cortisone values when compared to mean values of the three saliva samples.

Studies described in the previous paragraphs are summarised in Table 3. They are classified by species, and the analytical method, type of stressor, and values obtained are indicated.

## 5. Studies Where Cortisol and Corticosterone Were Measured in Combination

### 5.1. Studies on Animals

#### 5.1.1. Studies on Cows

Cortisol and corticosterone concentrations in cow serum after the application of different stress models were evaluated by spectrophotometry, RIA, and EIA [78,79]. These models were ACTH injection and intramammary bacterial infection, respectively. One hour after ATCH injection there was a more than two-fold increase in cortisol levels, decreasing again two hours post-injection, while corticosterone levels remained in similar concentrations before and after injection [78]. Similarly, in the second model [79], cortisol showed higher increases than corticosterone. A positive correlation between cortisol levels and increased rectal temperature was also observed (*r* ≈ 0.7).

#### 5.1.2. Studies on Birds

Cortisol and corticosterone concentrations were measured in plasma and feathers of sparrows by LC-MS/MS [80]. The feathers analysed were from birds at the time of the autumn moult, after the season of food abundance, and before winter stress. The analysis of plasma samples determined the presence of circulating corticosterone, but no cortisol levels were detected. Cortisol and corticosterone showed similar concentrations in feathers at the time of sampling, and an increase in both analytes in feathers was related to lower survival in the next winter season. The authors indicate that this difference between plasma and feather cortisol levels may be due to a localised secretion of cortisol in feather follicles or skin.

Cortisol and corticosterone concentrations were measured in plasma and different organs (bursa, thymus, spleen, and brain tissue) of starlings on the same day of hatching (P0) and ten days later (P10), before and after food restriction at each time [81]. The measurement was carried out by RIA for corticosterone and EIA for cortisol. At P0, no statistically significant differences were found in plasma and tissue cortisol and corticosterone before and after food restriction. At P10, there was a significant increase in corticosterone levels in plasma and all tissues analysed and a significant increase in cortisol levels in plasma, thymus, and brain after food restriction. However, in line with the previous report, plasma cortisol levels were significantly lower than corticosterone levels, with values below 2 ng/mL, whereas corticosterone values had a mean of 8.30 ng/mL in basal conditions. 

In addition, studies were carried out on plasma from farmed ducks to determine cortisol and corticosterone levels by EIA and RIA respectively after transport and ACTH injection [82,83]. In both cases, corticosterone levels were higher than cortisol at baseline and showed a greater increase (up to 4.55-fold) after the stressor. However, cortisol concentrations showed similar dynamics and a good correlation with corticosterone levels, so the authors consider it as an alternative to assess acute stress in this species.

#### 5.1.3. Studies on Laboratory Rodents

Overall, rodents are considered corticosterone dominant species, with cortisol levels being <1% of corticosterone levels [84]; however, there are rodent species such as squirrels in which cortisol concentrations were found to be equal to or even higher than corticosterone [85,86].

In mice, corticosterone and cortisol were measured by EIA and RIA, respectively, to assess the response to acute (48 h of uninterrupted movement restriction and forced swimming) and chronic (movement restriction 8 h/day for 23 days) stress [87]. During acute stress, cortisol levels increased earlier and remained increased? longer than corticosterone levels. When chronic stress was applied, while cortisol did not show any significant change, corticosterone levels decreased significantly from day 1 onwards. 

In hamsters, corticosterone and cortisol in serum were measured by RIA after chronic restrictive stress and acute stress [88]. At basal time, corticosterone levels were higher than cortisol levels. Following the acute stressor, the concentration of both glucocorticoids increased, with corticosterone levels being higher than cortisol levels. However, after chronic stress, there was only an increase in cortisol values. 

In other rodent species such as tuco-tucos, an increase in cortisol after acute stress was observed but corticosterone concentrations were not increased [86]. 

These data suggest that both hormones are independently regulated and that react differently, possibly due to differences in the sensitivity of each glucocorticoid to the hormone ACTH, depending on the species and the stressor, which is consistent with other studies [89]. The high variability in the response leads to recommending the measurement of both corticosteroids to assess adrenal function in rodents.

#### 5.1.4. Studies on Other Species

In amphibians, water-borne cortisol and corticosterone produced by captive-bred *Rana berlandieri* tadpoles were measured using an EIA [90]. An ACTH injection was used as a stress model to compare both glucocorticoids. While cortisol release decreased after ACTH injection, corticosterone levels in water after injection increased significantly. Therefore, the authors considered that corticosterone reflects the stress response better than cortisol in this experiment.

### 5.2. Studies on Humans

Corticosterone circulates in the blood at levels 10–20 times lower than cortisol in humans [91]. Therefore, it is not common to find studies that measure this glucocorticoid. However, plasma corticosterone concentrations using an EIA after intense exercise were determined [70], showing a similar increase to cortisol.

Studies described at this point are summarised in Table 4. They are classified by species, and the analytical method, type of stressor, and values obtained are indicated.

## 6. Studies Where Total Steroids and Selected Glucocorticoids Were Measured in Combination

The measurement of faecal glucocorticoid metabolites to assess animal welfare, as an alternative to plasma, is gaining importance in recent years, especially in wildlife. This is due to the greater ease of sample collection, avoiding stress in animals, and the fact that samples are less affected by daily variations [11,92]. In this point, we will differentiate the studies carried out on cortisol-dominant species and those where the species are corticosterone-dominant.

### 6.1. Studies on Cortisol-Dominant Species

Assays that were carried out for faecal glucocorticoid metabolite measurements in cortisol-dominant species such as elephants, antelopes, tigers, and primates are based on EIAs whose antibody has an affinity for the metabolite 11β-hydroxyetiocholanolone, recognising cortisol metabolites with a 5ß-reduced structure [54,93,94,95].

Different studies have compared total steroid and cortisol values, obtaining diverse results. Some studies in bears and monkeys in models of stress consisting of an ACTH injection carried out several EIAs in faeces for different cortisol metabolites and also measured cortisol. In these cases, the cortisol assay showed a higher increase after the stressor with less variation between baseline concentrations [96,97]. However, an EIA against 11β-hydroxyetiocholanolone was the most sensitive to detect stress after a model consisting of routine management in mandrills [98].

### 6.2. Studies on Corticosterone-Dominant Species

In a study in rodents, plasma corticosterone concentrations were compared with those of corticosterone metabolites in faeces, measured by two different EIAs (a 5a-pregnane-3b,11b,21-triol20-one EIA and an 11-oxoetiocholanolone EIA) after a stressor [99]. The 5a-pregnane-3b,11b,21-triol20-one EIA was the most sensitive to detect stress in this model.

## 7. Conclusions

There are three main glucocorticoids used to evaluate stress: cortisol, cortisone, and corticosterone, which vary in their amounts in different sample types and animal species. In addition, there is the concept of total steroids, which is used when immunoassays with antibodies showing cross-reactivity with different glucocorticoids or related metabolites are employed, being mostly used in faeces.

The examination of the reports included in this review, in which a variety of these glucocorticoid types are measured together, gives two ideas that can be used for future studies to assess animal stress or welfare status. One is the possibility of using a variety of these glucocorticoids in combination, providing on some occasions more information than assessing a single type. For example, the measurement of both cortisol and cortisone in mammals allows the evaluation of the activity of 11β-hydroxysteroid dehydrogenase in the case of using saliva or hair as a sample. The second is that, since the behaviours of these specific glucocorticoids vary depending on the species and the stressor stimulus, it would be recommend to do pilot studies to elucidate which glucocorticoid/s could be more appropriate to be evaluated, if no previous references are available.

## Figures and Tables

**Figure 1 metabolites-13-00106-f001:**
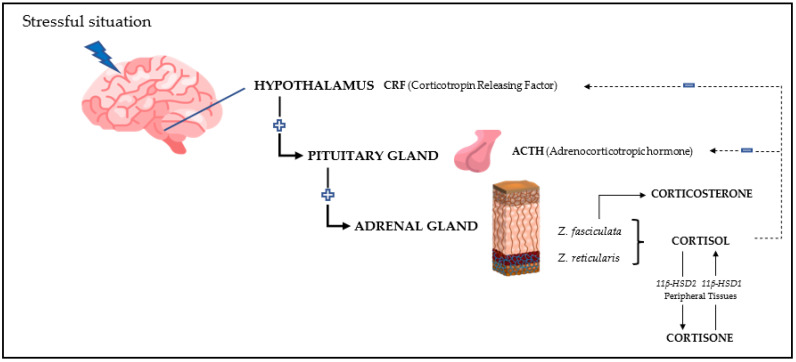
Schematic representation of glucocorticoid release following a stressful situation.

**Table 1 metabolites-13-00106-t001:** Main glucocorticoids and their main characteristics.

	Cortisol	Cortisone	Corticosterone
Formula	11β,17α,21-trihydroxypregn-4-ene-3,20-dione [17] ^1^ 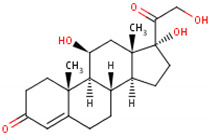	17-hydroxy-11-dehydrocorticosterone [18] ^2^ 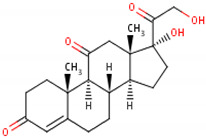	11β,21-dihydroxypregn-4-ene-3,20-dione [19] ^3^ 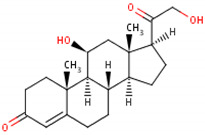
Structural differences	An extra hydroxyl group attached to the 17th carbon [20].	A ketone group attached to the 17th carbon [21].	No extra hydroxyl group on the 17th carbon [20].
Metabolism	Synthesised from pregnenolone in adrenal gland. Inactivated mainly in the kidney by 11β-hydroxysteroid dehydrogenase (11β-HSD) type 2 into cortisone [22,23].	Transformation in the liver, lungs, ovaries, and central nervous system by 11β-HSD type 1 into cortisol [24].	Derived from pregnenolone in adrenal gland [19].
Activity	Active molecule [25]	Inactive molecule	Active molecule
Half-life	In plasma: 66 min In tissues: 12 h [26,27]	In plasma: 90 min [21]	In plasma: 60–90 min [28]
Predominant species	It is the main glucocorticoid in most mammals [29]	Same species as cortisol	It is the main glucocorticoid in rats, mice, birds, and reptiles, due to a lack of the enzyme 17-α hydroxylase [9]

^1^ Cortisol. (s.f.). ChEBI. https://www.ebi.ac.uk/chebi/searchId.do?chebiId=CHEBI:17650 (accessed on 15 November 2022); ^2^ Cortisone. (s.f.). ChEBI. https://www.ebi.ac.uk/chebi/searchId.do?chebiId=CHEBI:16962 (accessed on 15 November 2022); and ^3^ Corticosterone. (s.f.). ChEBI. https://www.ebi.ac.uk/chebi/searchId.do?chebiId=CHEBI:16827 (accessed on 15 November 2022).

**Table 2 metabolites-13-00106-t002:** Main methods used for glucocorticoid measurement.

Analyte	Analytical Method	Reference
Cortisol	EIA	[34,39,40]
	RIA	[41,42]
	Chemiluminescence	[43]
	AlphaLISA	[44]
	HPLC	[35,36]
	LC-MS/MS	[37,38]
Cortisone	AlphaLISA	[44]
	UHPLC-MS/MS	[45,46]
	LC-MS/MS	[47]
	LC-MS3	[48,49]
Corticosterone	EIA	[50,51]
	RIA	[52,53]
Total steroids	EIA	[54,55,56]
	RIA	[57,58]

**Table 3 metabolites-13-00106-t003:** Examples where cortisol and cortisone were measured in combination after a stressor.

Species	Study	Cohort	Analytical Method	Stressor	Matrix	Values (Plasma/Saliva: ng/mL; Hair: pg/mg; Water-Borne: ng/L^−1^)		
						Metabolite	Before Stressor	After Stressor
Pig	[63]	14	LC-MS/MS	Nasal snare	Saliva (Sl) Plasma (P)	Cortisol	Sl: 0.06–0.25 * P: 100 *	Sl: 1–4 * P: 60–140 *
						Cortisone	Sl: 0.01–0.125 * P: 19 *	Sl: 0.25-1 * P: 17–33 *
	[44]	32	AlphaLISA	Farrowing	Hair	Cortisol	31.9	33.7
						Cortisone	119.9	527.2
	[62]	25	LC-MS/MS	Catheterisation	Serum (S)	Cortisol	42.8	71
						Cortisone	1.8	19
Human	[71]	197	LC/MS	Emotional stress	Hair	Cortisol	3.2	3.7
						Cortisone	5.9	7.4
	[49]	239	HPLC	Pregnancy	Hair	Cortisol	ND	3.75
						Cortisone	ND	14
	[72]	229	LC/MS	Ocean-going fishing 1–3 months	Hair	Cortisol	12.8	10.5
						Cortisone	3.3	4.9
	[67]	67	LC-MS/MS	Trier Social Stress Test	Saliva Serum	Cortisol	S: 2.2 Sl: 0.7	S: 17.5 Sl: 0.41
						Cortisone	4.2	Sl: 9
	[70]	12	EIA (cortisol) Chemiluminescence (cortisone)	GXT (morning)	Plasma Saliva	Cortisol	P: 170 Sl: 2.6	P: 250 Sl: 4.9
						Cortisone	P: 37.5 Sl: 13.6	P: 72.1 Sl: 21.1
Others species	[64]	120	LC-MS/MS	Air exposure	Plasma	Cortisol	10 *	55 *
						Cortisone	10 *	40 *
	[65]	12	RIA	Air exposure	Water-borne	Cortisol	1.1	25.2
						Cortisone	0.7	8 *
	[66]	24	EIA	Bacterial inoculation	Hair	Cortisol	9 *	2 *
						Cortisone	100 *	170 *

* Approximately (based on the graph presented in the referenced article).

**Table 4 metabolites-13-00106-t004:** Examples where cortisol and corticosterone were measured in combination after a stressor.

Species	Study	Cohort (n)	Analytical Method	Stressor	Matrix	Values (Plasma/Saliva: ng/mL; Faces/Tissues: ng/g; Feathers: ng/g; Water-Borne: pg/g)		
						Metabolite	Before Stressor	After Stressor
Cow	[78]	18	IDMS (Isotope dilution and spectrophotometry)	Injection of ACTH	Serum	Cortisol	3–6	4.1–8.9
						Corticosterone	2.4–3.5	3–4.1
	[79]	10	RIA (cortisol) EIA (corticosterone)	LPS infection	Plasma (P)	Cortisol	0.5	18
						Corticosterone	0.4	2.8
Birds	[80]		LC-MS/MS	Moult	Plasma Feathers (F)	Cortisol	P: 0.17	P: 0 F: 4.4–75.5
						Corticosterone	P: 8.6	P: 13–17 F: 4.1–372.9
	[81]	70	EIA (cortisol) RIA (corticosterone)	Restraining (P10)	Plasma Tissues (T)	Cortisol	P: 0.9 * T: 0.5–1.5 *	P: 1.5 * T: 1–2.2 *
						Corticosterone	P: 11 * T: 2–8 *	P: 30 * T: 5–20 *
Rodents	[88]	Not specified	RIA	Acute (A): supine restraint Chronic (C): cold restraint (2–4/day)	Plasma	Cortisol	4 *	A: 60 * C: 15 *
						Corticosterone	12 *	A: 65 * C: 15 *
	[87]	6	RIA (cortisol) EIA (corticosterone)	Acute: restraint, forced swimming Chronic: restraint	Serum	Cortisol	8–14 *	A: 30–35 * C: 15–30 *
						Corticosterone	40–160 *	A: 800–1200 * C: 200–1000 *
Others species	[90]	17 tadpoles	EIA	ACTH injection	Water-borne	Cortisol	4–5 *	3–3.5 *
						Corticosterone	100–180 *	180–350 *
Human	[70]	12	EIA	Graded exercise test	Plasma	Cortisol	170 *	250 *
						Corticosterone	14 *	47 *

* Approximately (based on the graph presented in the referenced article).

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
