# Peer review of "Different Types of Glucocorticoids to Evaluate Stress and Welfare in Animals and Humans: General Concepts and Examples of Combined Use"

_metabolites, 2023, doi:10.3390/metabo13010106_

Round 1
Reviewer 1 Report
In this paper, the authors described in an intriguing way the main glucocorticoids involved in stress response, namely, cortisol and cortisone in most
mammals, and corticosterone in birds and rodents. Indeed, they addressed the characteristics of the main glucocorticoids used to assess stress, as well as the main techniques and samples used for their quantification. In addition, the authors analyzed the studies where at least two of the main glucocorticoids have been measured in combination. In their conclusions, the authors speculated that, based on the different behavior of the main glucocorticoids depending on the species, among other factors, the combined measurement of at least two different types of glucocorticoids might provide more information on the welfare status of the animal than a single analyte.
This narrative paper is well-written and worth to be published in MDPI Metabolites.
I do suggest:
- to improve the English quality of the paper
- to clearly underline the “real” novelty of this review since only in PubMed a reader may find dozens of papers dealing with similar goals
- to present pictures schematically describing the connection between such glucocorticoids and stress
- an updated definition of “stress” should be included
Reviewer 2 Report
The review by Botía et al. (metabolites-2103800) “Different types of glucocorticoids to evaluate welfare in animals and humans: general concepts and examples of combined use” describes the general characteristics of glucocorticoids used in stress responses, and the main techniques for their quantification. The authors additionally added an interesting comparative analysis of the effect of the combination of at least two types of glucocorticoids (cortisol and cortisone; cortisol and corticosterone; steroids and glucocorticoids).
I have enjoyed reading this review, most clearly written. The manuscript provides updated and logically structured information from recent work on this intriguing field and is of great current interest. This review could assist in understanding the knowledge that the scientific community already acquired on the impact of glucocorticoids on the welfare of animals and humans.
I hope that my following comments can help to get an improved version:
Major comments:
- Taking into consideration that the topic is being widely studied and that this is a Review, I consider that the references to studies on the subject could be expanded. I found some articles that might be added if the authors consider them appropriate:
o The usefulness of measuring glucocorticoids for assessing animal welfare. Journal of Animal Science, Volume 94, Issue 2, February 2016, Pages 457–470, https://doi.org/10.2527/jas.2015-9645
o Manipulating glucocorticoids in wild animals: basic and applied perspectives. Conserv Physiol. 2015; 3(1): cov031. doi: 10.1093/conphys/cov031
o Review of human-animal interactions and their impact on animal productivity and welfare. Journal of Animal Science and Biotechnology volume 4, Article number: 25 (2013). https://doi.org/10.1186/2049-1891-4-25
- The definition of glucocorticoid is poorly addressed (line 49). This definition needs to be more widely described to fully understand the combined effect in the last sections of the Review. Please, consider introducing the term in section 1., and include a brief description of the molecular basis, receptor binding, etc., in one of the first two sections.
- What is considered “welfare” for the authors? Please, explain it and reference it, if applicable, in the Introduction (line 47).
- I missed a schematic Figure in 1. or 2. sections about how stressful stimulus triggers in the last instance the secretion of glucocorticoids. It could be a great help in the initial understanding of the definition of a glucocorticoid.
- When you analyze results in other species, could you clarify if it is in wild or farm animals? This is not always clear. The type of stress can be different attending on their relationship with humans and could interfere with the results. It would be interesting to compare the answer of the same species of a wild animal with the homologous on the farm.
- Are there experiments on domestic animals/pets such as dogs and cats?
- Conclusions: the considerations for future studies with glucocorticoids are vaguely explained.
Minor comments:
- Lines 22-24: could the authors rewrite this sentence? Too long and seems to repeat the idea of the previous ones. The end of the abstract could highlight the significant contribution of this review to the field.
- Lines 99-104: reference 56?
Round 2
Reviewer 1 Report
I found the paper quite improved and ready to be published after a moderate revision of the English